# Descriptive study of cattle and dog rabies cases in the Sheki-Zagatala region of Azerbaijan (2015–2016): Knowledge, attitudes, and practices of people towards rabies

Eldar Hasanov[1], Aytan Garayusifova[2][¤a]*, Eric Jon Tongren[3][¤b], Marika Geleishvili[3]

1 Azerbaijan Veterinary Research Institute, Baku, Azerbaijan, 2 School of Applied and Natural Sciences, Western Caspian University, Baku, Azerbaijan, 3 United States–Centers for Disease Control and Prevention Office, Tbilisi, Georgia

☯ These authors contributed equally to this work.
¤a Current address: Epidemiology Department, University of Washington, Seattle, Washington, United States of America
¤b Current address: Centers for Disease Control and Prevention Office, Addis Ababa, Ethiopia
* ayten.yusifzade@gmail.com

**Data Availability Statement:** The authors confirm that all data underlying the findings are fully available without restriction. The data underlying

## Abstract

Every year, rabies causes great damage to human health and the economy of countries around the world. This neurotropic and zoonotic viral disease is endemic to Azerbaijan. This study describes cattle and dog rabies cases identified between 2015 and 2016. In addition, a questionnaire was administered to 100 respondents comprised of case owners, and non-case animal owners, to assess knowledge, attitudes, and practices within this population. The study demonstrates a general lack of knowledge of rabies in the participating communities. The majority of respondents were familiar with rabies and understood that animal bites are a source of transmission. However, many respondents did not know that rabies is preventable and admitted not knowing additional routes of transmission. In addition, there was less perceived risk with contact with animal saliva. Despite free vaccinations in Azerbaijan, only 27 out of 45 dogs in this study were vaccinated. Although educational programming and preventive vaccination of dogs has been implemented, and significant progress has been made in the sphere of epidemiological surveillance and prevention, rabies cases remain problematic in the Sheki-Zagatala region. Regular educational programs for communities, strengthening of the existing vaccination programs, and a comprehensive epidemiological case-control study to identify the disease risk factors could help reduce the burden of rabies in Azerbaijan.

## Introduction

Rabies is a viral disease of the central nervous system caused by a virus belonging to the *Rhabdoviridae* family and *Lyssavirus* genus [1]. Infection of humans and animals occurs through direct contact with the virus: as a result of a bite or contact of saliva with damaged skin or

the results presented in the study are available within the paper.

**Funding:** HE and GA were funded as South Caucasus Field Epidemiology and Laboratory Training Program (FELTP) residents supported through the CDC Georgia office. As a representative of the funder office, GM played a role in the study design and data collection.

**Competing interests:** The authors have declared that no competing interests exist.

mucous membranes, other inoculation routes are rare [2,3]. The virus replicates in muscle tissue and travels up the central nervous system which causes infection of the brain (encephalitis and paralysis) [4]. Dogs are a major source of human rabies [5]. Rabid dog manifests the symptoms of hypersalivation, aggressiveness and biting, paralysis, difficulty walking or circular walk, ataxia, glassy eyes and difficulty breathing and swallowing [6]. Encephalitic (furious) or paralytic (dumb) clinical forms of rabies occur at an approximate ratio of 2:1 in both humans and dogs [5,7].

Human rabies remains a major public health problem in the world. Although effective human and animal vaccines are widely available for its prevention and control, each year about 10 million people receive post exposure rabies vaccination [8,9]. Globally, an estimated 59,000 human deaths are caused by rabies each year, especially in Asia and Africa with dogs causing > 99% of these cases. [3,10].

Despite the adoption and implementation of rabies elimination strategies by many countries, rabies remains problematic. Over 1 million animals die from rabies every year, causing enormous economic damage that is estimated at millions of US dollars annually [11,12]. Due to effects on livestock, the social and economic impacts of rabies is increasing. Also, the spread of rabies in wild animal populations is contributing to epizootics in farm animals, especially in cattle [13].

Although rabies is a notifiable disease in Azerbaijan and controlled by vaccination, cases continue to increase among people and animals. In recent years, the preventive vaccination of dogs and cats are carried out in a planned manner [14]. Between 2015 and 2016, approximately 465,000 animals were vaccinated against rabies (1,217,559 animals from 2012 through 2016). In addition, oral vaccination of wild animals was conducted in select forests and suburbs in 2015. However, there were 9 human rabies cases and 66 animal rabies cases from Jan 1, 2015 through Aug 14, 2016 registered throughout Azerbaijan. This includes 26 farm animals (large and small ruminants), 33 dogs, and 7 wild animals (5 jackals and 2 wolves) [15,16].

In 2013 and 2014, the Azerbaijan Ministry of Health and Ministry of Agriculture conducted educational programs in the Sheki-Zagatala region that focused on improving people's knowledge on rabies. These included information on rabies disease, routes of transmission, protective measures for humans and animals, and rabies vaccine efficacy. Because Sheki-Zagatala is proximal to forests and accompanying wildlife population, and because jackals were found positive for rabies, oral vaccination of wildlife was included in the 2015 vaccination program. However, from 2013–2016, 30% (42/139) of all animal cases in Azerbaijan still came from the Skeki-Zagatala region.

This study describes the epidemiological profile of animal rabies cases in the Sheki-Zagatala region from 2015–2016. The objectives of this study are to assess the community knowledge, attitudes, and practices of the residents of Sheki-Zagatala region and provide a case for the need of a larger national-level study to assess risk factors and provide further guidance to public health leaders in Azerbaijan.

## Materials and methods

### Study area

The study was conducted from August 11–19, 2016 in northwest region of Azerbaijan in Sheki-Zagatala. The region borders Russia on the north and Georgia on the West. Almost 27% of the region is covered by forest. Agriculture is the primary basis of the economy in this region, producing 9.1% of the country's agricultural products.

## Study design and data collection

The confirmed cattle and dog rabies cases in Sheki-Zagatala region occurring between Jan. 1, 2015 through Aug. 14, 2016 were taken from the Republican Veterinary Laboratory (RVL) database. Additional cases were classified as probable cases based on their epidemiological link to confirmed rabies cases (date of onset, common herd, common owners, common pastures, etc.) and clinical symptoms such as aggressiveness, anorexia, pupil enlargement, glassy eyes, hyperactivity, spumescent discharge.

No ethics or approval by an IRB was requested for this study. The study participants were informed about the study and verbal consent was obtained from all participants prior the interview. The abstracted/collected data was properly stored and secured. The questionnaire survey was conducted among 100 cattle and dog owners, including owners of confirmed rabid animals taken from RVL database, by face-to-face interview. The questionnaire was designed to collect demographic information and information about the respondents' knowledge, attitudes, and practices of rabies; treatment; and prevention practices.

The latitude and longitude of the animal rabies cases in this study were obtained using Google Maps. The GPS coordinates were used to construct a map using an open-source tool from OpenStreetMap.

## Results

### Characteristics of dog and cattle cases

We identified a total of 20 dog and cattle rabies cases. Cases were comprised of an initial 14 cases extracted from RVL database (9 cattle/5 dogs) and supplemented by another 6 (2 cattle/4 dogs) probable cases identified during the study through epidemiological links, symptoms of death, and date of onset of disease.

### Cases among dogs

Table 1 describes the 9 dog cases where 6 roamed untethered in the community. None of the dogs had a registration number. Five of these 9 dogs were bitten, including 3 dogs that had been vaccinated. All 9 of the dogs died.

### Cases among cattle

Table 1 also describes the 11 cases in cattle; none had a registration number. Three cattle separately grazed alone. The other eight cases came from independent herds. Seven out of 11 cases were animals that grazed near forests. Nine of these 11 cattle were vaccinated against Anthrax, Lumpy Skin Disease (LSD) and Foot and Mouth Disease (FMD). Among eleven cases, seven cattle were bitten and all of them died. This includes 3 animals that were vaccinated after receiving bites.

In total, 12 of these 20 animal cases were confirmed to have been bitten. Four of these bites came from Jackals and 4 came from dogs. For the other 4 animals, the owners did not know what animal bit their animals, but wounds were consistent with bites (lesions on body, scratches, incisions, bruises etc.). Only 6 of the animals were vaccinated against rabies. Three dogs received pre-exposure prophylaxis (PrEP) rabies vaccination; 3 cattle got post-exposure prophylaxis (PEP) vaccination after being bitten. All of them have died.

The GPS locations of the 20 dog and cattle cases are shown in Fig 1. These identified cases represent 30% (20 out of 65) of all dog and cattle cases in Azerbaijan during this time period.

Of the 20-animal case-owners surveyed, observed symptoms were consistent with clinical diagnosis. The breakdown of clinical signs and symptoms by animal type is shown in Fig 2.

**Table 1. Description of cases of rabies in dogs and cattle.**

| Dogs | n = 9 | Cattle | n = 11 |
|---|---|---|---|
| **Animal registration number** | | **Animal registration number** | |
| Yes | 0 | Yes | 0 |
| No | 9 | No | 11 |
| **Age of the animal** | | **Age of the animal** | |
| 3–18 months | 2 | < 3 months | 1 |
| 18 months—4 years | 6 | 3–18 months | 3 |
| >4 years | 1 | 18 months—4 years | 3 |
| | | >4 years | 4 |
| **Dog kept condition** | | **Type of movement** | |
| Chained in the backyard | 1 | Herd-type | 7 |
| Is not chained, but never leaves the backyard | 2 | Seasonal movement | 3 |
| Always goes out from the backyard | 4 | Is not moving, always in the pen | 1 |
| Goes with cattle to the pasture | 1 | | |
| Hunting dog | 1 | | |
| | | **Type of pasture** | |
| | | Common | 6 |
| | | Surrounded | 3 |
| | | No pasture | 2 |
| | | **Livestock graze alone or together with other animals** | |
| | | Alone | 3 |
| | | With other animals | 8 |
| | | **Location of pastures** | |
| | | Near the forest | 7 |
| | | Near the open area-field | 4 |
| | | **Vaccination against any diseases** | |
| | | Yes | 9 |
| | | No | 2 |
| **Vaccination against rabies** | | **Vaccination against rabies** | |
| Yes | 3 | Yes | 3 |
| No | 6 | No | 8 |
| **Type of rabies vaccination** | | **Type of rabies vaccination** | |
| Mandatory | 0 | Mandatory | 3 |
| Preventive | 3 | Preventive | 0 |
| **Animal bitten by any animal** | | **Animal bitten by any animal** | |
| Yes | 5 | Yes | 7 |
| No | 2 | No | 1 |
| I don't remember | 2 | I don't remember | 3 |
| **Bitten by which animal?** | | **Bitten by which animal** | |
| Jackal | 1 | Jackal | 3 |
| Wolf | 0 | Wolf | 0 |
| Rat | 0 | Rat | 0 |
| Fox | 0 | Fox | 0 |
| Dog or Stray dog | 1 | Dog or Stray dog | 3 |
| Do not know | 3 | Do not know | 1 |
| **Geographic District (# of villages)** | | **Geographic District (# of villages)** | |
| Zagatala (9) | 3 | Zagatala (9) | 2 |

*(Continued)*

**Table 1.** (Continued)

| Dogs | n = 9 | Cattle | n = 11 |
|---|---|---|---|
| Sheki (14) | 3 | Sheki (14) | 3 |
| Gakh (16) | 2 | Gakh (16) | 6 |
| Balaken (3) | 1 | Balaken (3) | 0 |

Foamy mouth discharge (60%), an aversion to food (40%), and difficulty swallowing (35%) were among the most reported symptoms.

## Demographics of questionnaire respondents in the Sheki-Zagatala region

Among 100 questionnaire respondents, 69 were males. The ages for the majority of the respondents fell within the range of 30–49 (29 persons) and 50–69 (53 persons) years old. With respect to education level, 7 had completed primary education, which is about 3-years of school in Azerbaijan, and 78 attended or had not completed high school. Only 15 respondents went further for higher education (Table 2).

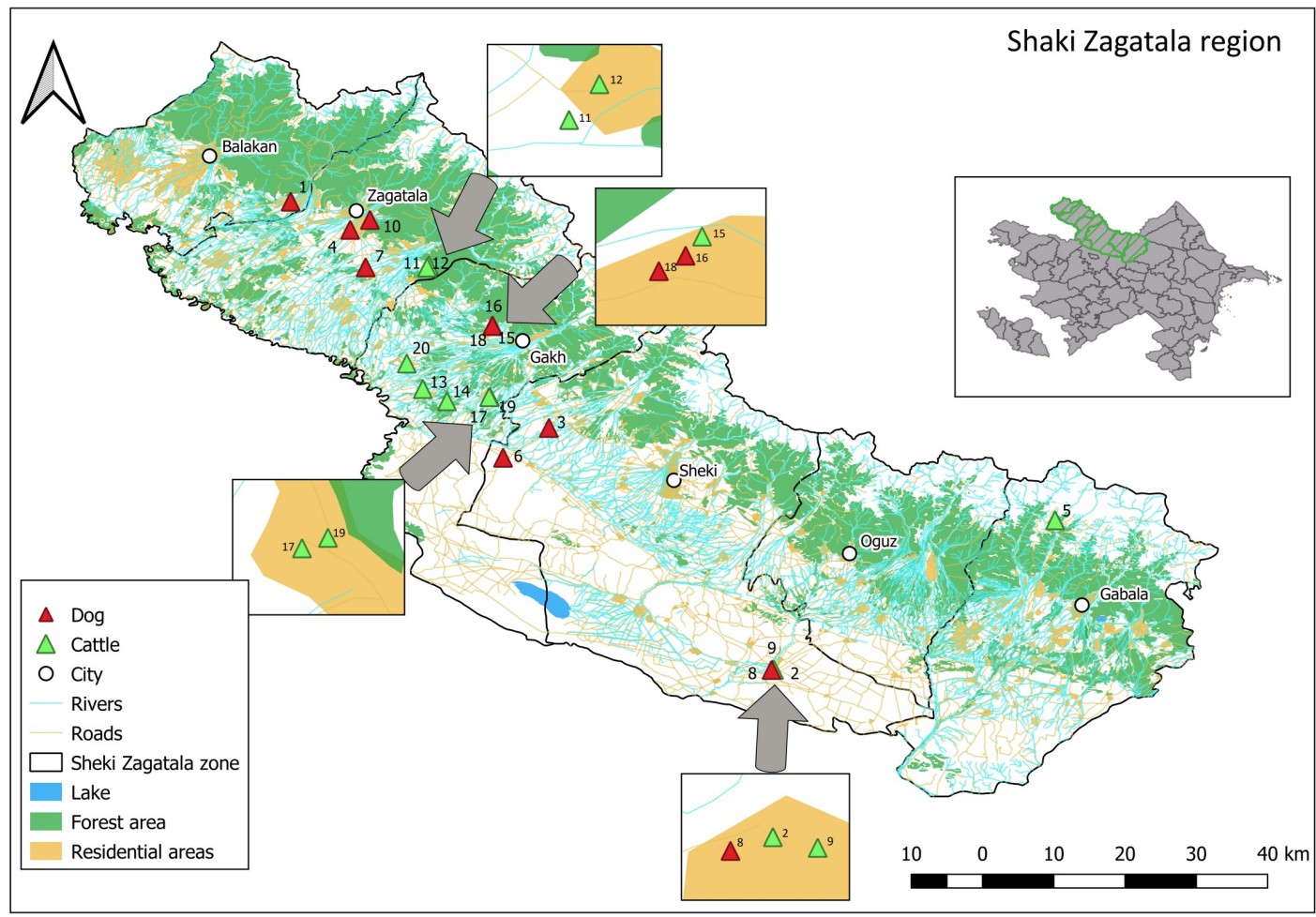

**Fig 1. Registered cattle and dog rabies cases in Sheki-Zagatala districts for 2015 and 2016.** Base map and data from OpenStreetMap and OpenStreetMap Foundation.

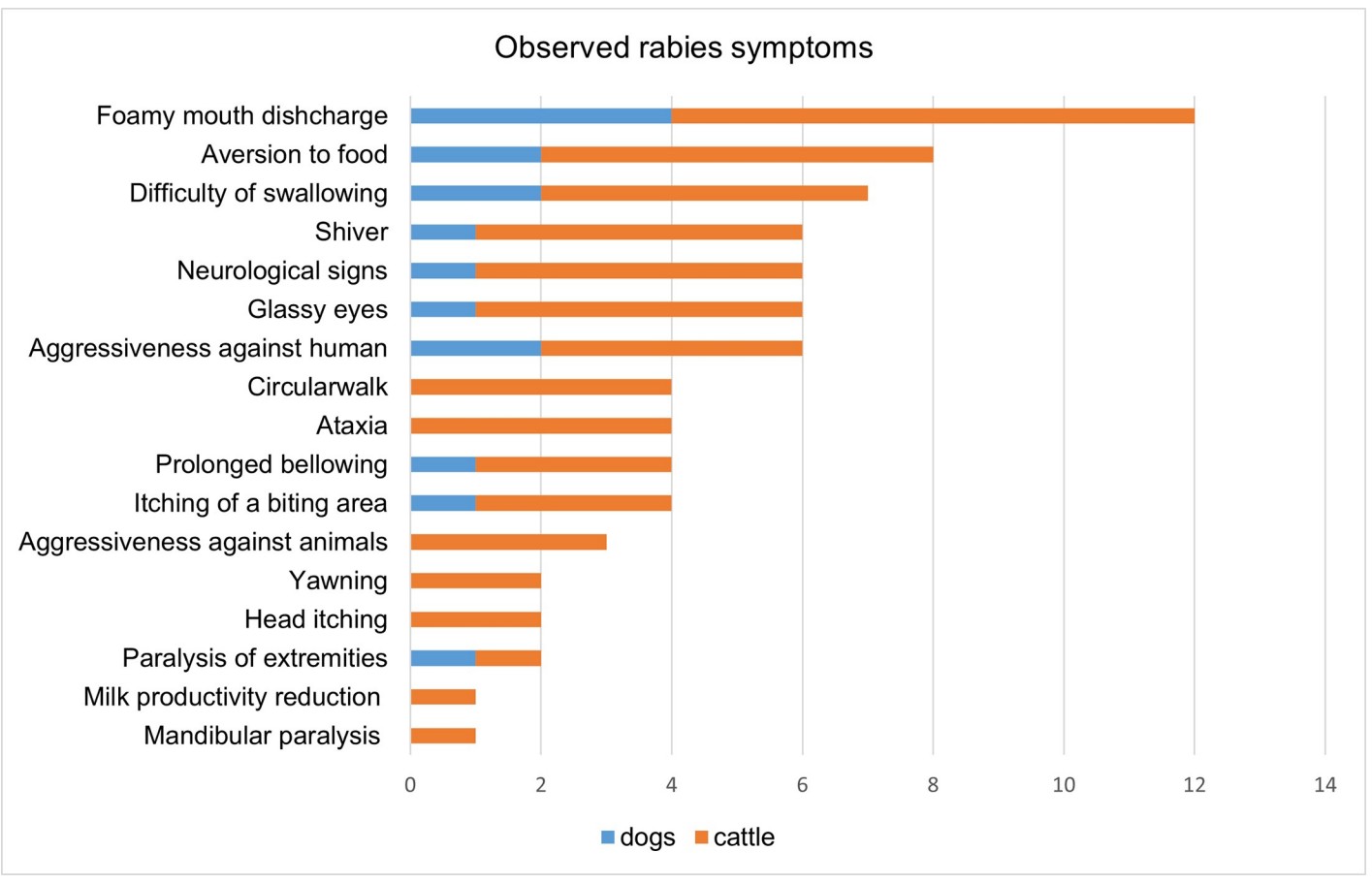

**Fig 2. Most common reported symptoms for rabid cattle and dogs by animal owners (N = 20).**

## Knowledge, attitudes, and practices of residents in the Sheki-Zagatala region

As indicated in Table 3, almost all respondents (94 out of 100) were familiar with the rabies disease. The majority were aware that dog bites (97) and cattle bites (91) are risks for exposure

**Table 2. Demographic characterization of respondents.**

| Characteristics | Respondents N = 100 |
|---|---|
| **Gender** | |
| Male | 69 |
| Female | 31 |
| **Age** | |
| <30 | 6 |
| 30–49 | 29 |
| 50–69 | 53 |
| >70 | 12 |
| **Education level** | |
| Elementary | 7 |
| High school (incomplete, complete) | 78 |
| Higher education (incomplete, complete) | 15 |

**Table 3. Assessment of knowledge, attitudes, and practices of the respondents on rabies (N = 100).**

| Question | Respondents |
|---|---|
| | **N = 100** |
| **Q1. Do you know what the disease rabies is?** | |
| Yes | 94 |
| No | 6 |
| **Q2. If a dog or cat bites you?** | |
| I'll go to the doctor, and if he advises, I'll be vaccinated | 97 |
| I will go to the doctor, but I will not be vaccinated, despite his advice | 1 |
| I don't pay attention; it happens so often | 0 |
| I will take care of the damage by myself at home, on my own | 2 |
| **Q3. If a cow bites you?** | |
| I'll go to the doctor, and if he advises, I'll be vaccinated | 91 |
| I will go to the doctor, but I will not be vaccinated, despite his advice | 0 |
| I don't pay attention; it happens so often | 4 |
| I will take care of the damage by myself at home, on my own | 5 |
| **Q4. If you have contact with the saliva of a dog or cat?** | |
| I'll go to the doctor, and if he advises, I'll be vaccinated | 60 |
| I will go to the doctor, but I will not be vaccinated, despite his advice | 0 |
| I don't pay attention; it happens so often | 16 |
| I will take care of the damage by myself at home, on my own | 24 |
| **Q5. If you have contact with the saliva of a cow?** | |
| I'll go to the doctor, and if he advises, I'll be vaccinated | 52 |
| I will go to the doctor, but I will not be vaccinated, despite his advice | 1 |
| I don't pay attention; it happens so often | 21 |
| I will take care of the damage by myself at home, on my own | 26 |
| **Q6. What animals may be infected by rabies?** | |
| Cattle | 85 |
| Sheep | 63 |
| Goat | 57 |
| Horse | 47 |
| Pig | 61 |
| Dog | 92 |
| Cat | 68 |
| Human | 80 |
| I don't know | 3 |
| **Q7. Is it possible to prevent animal from contracting rabies?** | |
| Yes | 74 |
| No | 14 |
| I don't know | 12 |
| **Q7a. If yes, how?** | |
| Vaccination | 64 |
| Keep them isolated | 10 |
| **Q8. In case of free vaccination, would you vaccinate your animals against rabies?** | |
| Yes | 94 |
| No | 6 |

(*Continued*)

**Table 3.** (Continued)

| Question | Respondents |
|---|---|
| | **N = 100** |
| **Q9. Will you vaccinate your animals against rabies if you have to pay for a vaccine?** | |
| Yes | 87 |
| No | 6 |
| I don't know | 7 |
| **Q10. How would you prefer to get information about rabies?** | |
| TV | 53 |
| Newspaper/magazines | 12 |
| From the vet | 66 |
| Through Consulting Center | 5 |
| Other | 15 (books, internet, neighbor, during practice, etc.) |

to the rabies virus and would attend to the doctor and be vaccinated if advised. The majority (80) of respondents are aware that human and domestic animals can be affected by rabies. Most animal owners understand that rabies is preventable through vaccination but only 64 indicated a preference of vaccination over animal isolation.

Respondents indicated a preference to receiving information and knowledge about rabies from veterinarians (66) and by television programs on rabies (53). Fewer respondents indicated a preference for learning about rabies through newspapers, consulting centers, or internet resources.

## Knowledge on rabies transmission

Given a multiple-choice question (N = 100) on the transmission of rabies, respondents chose bites (n = 96), saliva (83), scratch (67), abrasion (37) and skin contact (25) as the main modes of transmission for rabies (Table 4). Few cattle owners considered milking, consumption of milk or dairy, and consumption of meat (3) as a mode of transmission from large ruminants. Similarly, few animal owners considered rabies transmission through air droplets, urine, and feces (7). In total, 21 respondents indicated that they did not know how rabies is transmitted.

**Table 4. Respondent knowledge about rabies transmission (N = 100).**

| Transmission ways | Transmitted from cattle | Transmitted from dogs |
|---|---|---|
| Bites | 20 | 76 |
| Saliva | 54 | 29 |
| Scratch | 40 | 27 |
| Abrasion | 20 | 17 |
| Skin contact | 17 | 8 |
| Milking | 5 | N/A |
| Consumption of milk or dairy | 3 | N/A |
| Consumption of meat | 3 | N/A |
| Contact with wild animal (or sick and dead wild animals) | 2 | 5 |
| Air droplets | 3 | 2 |
| Feces | 0 | 1 |
| Urine | 0 | 1 |
| Do not know | 10 | 11 |

## Discussion

Based on the questionnaire, it is clear that study participants from the Sheki-Zagatala region lack specific knowledge on rabies transmission, risk factors, and perceived seriousness of the disease. Bites and contact with animal saliva were correctly indicated as a means of transmission of the disease by a majority of respondents. However, there is less perceived risk associated with contact of dog and cattle saliva. In addition, 26 respondents did not believe, or understand, that animal rabies is preventable. Another 22 respondents indicated that they did not fully understand how rabies is transmitted.

The Sheki-Zagatala region is known for its proximity to the forest and wildlife. Based on RVL data domestic animals such as dogs, cats, cattle, and wild animals such as jackals and wolves are a common source and cause of rabies in Azerbaijan [14]. Respondents indicated these animals, together with pigs, horses, sheep, and goats as a cause of rabies. Despite the fact that pigs are not common in Azerbaijan, they were mentioned by 61 respondents as source of rabies.

Three rabies case-owners within this study (1 dog, 3 cattle) claimed that jackals were responsible for their animals' deaths. The RVL also confirmed rabies in 7 jackals and two wolves within 2015–2016. In addition, 13 respondents confirmed seeing dead jackals next to the pastures within the study area. Furthermore, 35 respondents saw wild animals near the pastures, and 63 saw wild animals near the villages. The location of pastures near forests was reported as risk factor by a previously conducted study in Azerbaijan [17]. A future and larger case-control study aimed at assessing the added risk of wild animals on rabies cases in Azerbaijan would be warranted. The study might compare this risk in the Skeki-Zagatala region to other regions in Azerbaijan.

The Azerbaijan Ministry of Agriculture provides free rabies vaccinations throughout the country for dogs and cats registered by veterinarians. Annual vaccinations are administered to dogs which require three vaccinations over a three-year period. The process involves veterinarians visiting each household with registered animals. However, owners are responsible for registering their animals. Despite such facilities, only 27 (60%) of the dogs in this study had preventative vaccination. Interestingly, 3 of the dogs that received a preventative vaccination were identified as case dogs and died. However, there was perceived recall bias associated with vaccination of the study animals. Three dog owners claimed that their vaccinated dogs were bitten and died. One dog was bitten by a stray dog and died without developing any rabies related symptoms; a second dog had scratches, he developed aggressiveness to owner, glassy eyes, aversion to food, foamy month, difficulty swallowing, shiver, and died; a third dog had scratches and the owner did not follow later developments, the animal was found dead. The respondents were asked to specify the exact date of the rabies vaccination of their dogs, but none of them could answer. The veterinarians were contacted in order to clarify the date of administration of pre-exposure prophylaxis rabies vaccination to case dogs. The veterinarians clarified that none of the dogs had completed their full vaccination programs and that rabies vaccinations were not current. Additionally, three cattle in the study received post-exposure prophylaxis rabies vaccine after being bitten by jackals (2) and a dog (1), but all died. All three developed some precent of rabies related symptoms–neurological, aggressiveness to owner, aversion to food, glassy eyes, foamy month, difficulty swallowing, shiver, ataxia, head itching, itching, and walking in circular patterns. The problem could be either late administration of the vaccine (after 72 hours from being bitten) or there was a lethal bite to the head. The owners didn't clarify these points. We suspect that vaccine-related problems in cattle and dogs are related to gaps in knowledge. Veterinarians are also responsible for providing enough

information to animal owners on how frequently and when to vaccinate their animals. Thus, there is also a need for a strong program.

In addition, this study revealed 6 probable cases that went unreported, identified through epidemiological links, symptoms of death, and date of onset of disease. We hypothesize that this is a result of a low overall level of knowledge about the disease, low perceived severity of the disease, and may also be due to limited access to testing, which is 400km away from Sheki-Zagatala. As such, we suspect that rabies cases are underreported in Azerbaijan.

Through the current study we were able to learn about issues that may be impacting rabies cases in Azerbaijan. These issues include access to veterinary care, the potential role of wild animal populations (7 rabid jackals and wolves confirmed by the RVL; respondents' observations of dead wild animals near pastures), incomplete vaccination program, knowledge on transmission, and deficiencies in knowledge about the signs and symptoms of rabies. However, we cannot make any generalization or come up with decision on spread of rabies with the limited number of animals (20) and the limited number of people (100) involved in this study. Despite preventive measures conducted in this region, including a dog vaccination program, rabies cases continue to occur in Sheki-Zagatala. Generally, there is an inadequate level of awareness about rabies and insufficient adoption of rabies control measures among study participants. Regular educational programs for this community would help. The vaccination program could also be strengthened to better control the spread in both domestic and wild animals. Ultimately, a more extensive study that includes more people and more animals is needed to make fully assess peoples' knowledge about rabies and to identify risk-factors most relevant to the spread of rabies in Azerbaijan.

## Acknowledgments

We would like to express our sincere gratitude to Defense Threat Reduction Agency (DTRA) Biological Threat Reduction Program and Dr. David Garcia from Civilian Research and Development Foundation (CRDF) for the support provided us during the analysis of results and review of the manuscript.

## Author Contributions

**Conceptualization:** Eldar Hasanov.

**Data curation:** Eldar Hasanov, Aytan Garayusifova.

**Formal analysis:** Eldar Hasanov, Aytan Garayusifova, Marika Geleishvili.

**Funding acquisition:** Eric Jon Tongren.

**Investigation:** Eldar Hasanov, Aytan Garayusifova.

**Methodology:** Eldar Hasanov, Aytan Garayusifova, Marika Geleishvili.

**Project administration:** Eldar Hasanov, Eric Jon Tongren, Marika Geleishvili.

**Software:** Aytan Garayusifova.

**Supervision:** Eldar Hasanov, Eric Jon Tongren.

**Writing – original draft:** Aytan Garayusifova.

**Writing – review & editing:** Eldar Hasanov, Aytan Garayusifova.

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
