## [Decision Letter · Decision Letter 0]

2 Oct 2020

PONE-D-20-26983

Descriptive study of animal rabies cases in Sheki-Zagatala region of Azerbaijan for 2015-2016

PLOS ONE

Dear Dr. Garayusifova

Thank you for submitting your manuscript to PLOS ONE. After careful consideration, we feel that it has merit but does not fully meet PLOS ONE’s publication criteria as it currently stands. Therefore, we invite you to submit a revised version of the manuscript that addresses the points raised during the review process.

Many thanks for submitting your manuscript to PLOS One

Your manuscript was reviewed by two experts in the field, who have recommended some changes be made prior to acceptance

I therefore invite you to make these changes, and write a response to reviewers to expedite re-review when you resubmit.

I wish you the best of luck with your revisions

Hope you are keeping safe and well in these difficult times

Thanks

Simon

We look forward to receiving your revised manuscript.

Kind regards,

Simon Clegg, PhD

Academic Editor

PLOS ONE

2. We note you have included a table to which you do not refer in the text of your manuscript. Please ensure that you refer to Table 4 in your text; if accepted, production will need this reference to link the reader to the Table.

3.We note that [Figure(s) 1] in your submission contain [map/satellite] images which may be copyrighted. All PLOS content is published under the Creative Commons Attribution License (CC BY 4.0), which means that the manuscript, images, and Supporting Information files will be freely available online, and any third party is permitted to access, download, copy, distribute, and use these materials in any way, even commercially, with proper attribution. For these reasons, we cannot publish previously copyrighted maps or satellite images created using proprietary data, such as Google software (Google Maps, Street View, and Earth). For more information, see our copyright guidelines: http://journals.plos.org/plosone/s/licenses-and-copyright.

1.    You may seek permission from the original copyright holder of Figure(s) [1] to publish the content specifically under the CC BY 4.0 license. 

Reviewers' comments:

Reviewer's Responses to Questions

**Comments to the Author**

1. Is the manuscript technically sound, and do the data support the conclusions?

Reviewer #1: Partly

Reviewer #2: Yes

2. Has the statistical analysis been performed appropriately and rigorously? 

Reviewer #1: No

Reviewer #2: N/A

3. Have the authors made all data underlying the findings in their manuscript fully available?

Reviewer #1: No

Reviewer #2: Yes

4. Is the manuscript presented in an intelligible fashion and written in standard English?

Reviewer #1: Yes

Reviewer #2: Yes

5. Review Comments to the Author

Reviewer #1: In addition to the review comments attached, the authors have selected two groups for interviewing. If the authors intend to compare the two groups then why haven't they performed a statistical analysis for the same? If the intention is not to compare the two groups then the groups can be merged.

Reviewer #2: The manuscript provides important information about rabies in Azerbaijan and authors suggested the further action for controlling rabies in the countries, however the manuscript needs to be improved before considering its publication.

6. PLOS authors have the option to publish the peer review history of their article (what does this mean?). If published, this will include your full peer review and any attached files.

Reviewer #2: No

---

## [Author Response · Author response to Decision Letter 0]

12 Feb 2021

1. Ensure manuscript meets PLOS ONE’s style requirements, including those for file naming

• Thank you for helping us ensure we have the correct formatting and style elements. We have updated the manuscript to ensure that it meets all style requirements for each section. 

2. Table 4 was not referenced in the text of the manuscript

• Thank you for catching this. We discussed the table but failed to reference the table. We have updated the manuscript to include a reference to the table in our discussion (page 18, line 204 in Revised Manuscript with Track Changes).

3. The map contains images which may be copyrighted. 

• Thank you for helping us understand this issue and for the helpful tips on developing a new map. We have generated a new map using an open-source tool from OpenStreetMap®. Documentation is licensed under the Creative Commons Attribution-ShareAlike 2.0 license (CC BY-SA 2.0). As such, we have the credited OpenStreetMap as and its contributors as required in the “Study design and data collection” subsection of the Materials and Methods (Page 7, Line 127-128).

---

## [Decision Letter · Decision Letter 1]

2 Mar 2021

Descriptive study of cattle and dog rabies cases in the Sheki-Zagatala region of Azerbaijan (2015-2016): Knowledge, attitudes, and practices of people towards rabies

PONE-D-20-26983R1

Dear Dr. Garayusifova,

We’re pleased to inform you that your manuscript has been judged scientifically suitable for publication and will be formally accepted for publication once it meets all outstanding technical requirements.

Kind regards,

Simon Clegg, PhD

Academic Editor

PLOS ONE

Additional Editor Comments:

Many thanks for resubmitting your manuscript to PLOS One

As you have addressed all the comments and the manuscript reads well, I have recommended it for publication

You should hear from the Editorial Office shortly.

It was a pleasure working with you and I wish you the best of luck for your future research

Hope you are keeping safe and well in these difficult times

Thanks

Simon

Reviewers' comments:

Reviewer's Responses to Questions

**Comments to the Author**

1. If the authors have adequately addressed your comments raised in a previous round of review and you feel that this manuscript is now acceptable for publication, you may indicate that here to bypass the “Comments to the Author” section, enter your conflict of interest statement in the “Confidential to Editor” section, and submit your "Accept" recommendation.

Reviewer #2: All comments have been addressed

2. Is the manuscript technically sound, and do the data support the conclusions?

Reviewer #2: Yes

3. Has the statistical analysis been performed appropriately and rigorously? 

Reviewer #2: N/A

4. Have the authors made all data underlying the findings in their manuscript fully available?

Reviewer #2: Yes

5. Is the manuscript presented in an intelligible fashion and written in standard English?

Reviewer #2: Yes

6. Review Comments to the Author

Reviewer #2: The authors revised the manuscript according to the comments made in the previous version.

The current version clearly provide the important findings according to the aims of the study as well as the limitations.

7. PLOS authors have the option to publish the peer review history of their article (what does this mean?). If published, this will include your full peer review and any attached files.

Reviewer #2: No

---

## [Editor Report · Acceptance letter]

8 Nov 2021

PONE-D-20-26983R1 

Descriptive study of cattle and dog rabies cases in the Sheki-Zagatala region of Azerbaijan (2015-2016): Knowledge, attitudes, and practices of people towards rabies 

Dear Dr. Garayusifova:

I'm pleased to inform you that your manuscript has been deemed suitable for publication in PLOS ONE. Congratulations! Your manuscript is now with our production department. 

Kind regards, 

on behalf of

Dr. Simon Clegg 

Academic Editor

PLOS ONE